# Soft Neighbors are Positive Supporters in Contrastive Visual Representation Learning

**Chongjian Ge**[1]    **Jiangliu Wang**[2]    **Zhan Tong**[3]    **Shoufa Chen**[1]    **Yibing Song**[4*]    **Ping Luo**[1]
[1]The University of Hong Kong    [2]The Chinese University of Hong Kong
[3]Tencent AI Lab    [4]AI[3] Institute, Fudan University
rhettgee@connect.hku.hk    jlwang@cuhk.edu.hk    zhantong.2023@gmail.com
shoufach@connect.hku.hk    yibingsong.cv@gmail.com    pluo@cs.hku.hk

## Abstract

Contrastive learning methods train visual encoders by comparing views (e.g., often created via a group of data augmentations on the same instance) from one instance to others. Typically, the views created from one instance are set as positive, while views from other instances are negative. This binary instance discrimination is studied extensively to improve feature representations in self-supervised learning. In this paper, we rethink the instance discrimination framework and find the binary instance labeling insufficient to measure correlations between different samples. For an intuitive example, given a random image instance, there may exist other images in a mini-batch whose content meanings are the same (i.e., belonging to the same category) or partially related (i.e., belonging to a similar category). How to treat the images that correlate similarly to the current image instance leaves an un-explored problem. We thus propose to support the current image by exploring other correlated instances (i.e., soft neighbors). We first carefully cultivate a candidate neighbor set, which will be further utilized to explore the highly-correlated instances. A cross-attention module is then introduced to predict the correlation score (denoted as positiveness) of other correlated instances with respect to the current one. The positiveness score quantitatively measures the positive support from each correlated instance, and is encoded into the objective for pretext training. To this end, our proposed method benefits in discriminating uncorrelated instances while absorbing correlated instances for SSL. We evaluate our soft neighbor contrastive learning method (SNCLR) on standard visual recognition benchmarks, including image classification, object detection, and instance segmentation. The state-of-the-art recognition performance shows that SNCLR is effective in improving feature representations from both ViT and CNN encoders.

## 1 Introduction

Visual representations are fundamental to recognition performance. Compared to the supervised learning design, self-supervised learning (SSL) is capable of leveraging large-scale images for pretext learning without annotations. Meanwhile, the feature representations via SSL are more generalizable to benefit downstream recognition scenarios (Grill et al., 2020; Chen et al., 2021; Xie et al., 2021; He et al., 2022; Wang et al., 2021). Among SSL methods, contrastive learning (CLR) receives extensive studies. In a CLR framework, training samples are utilized to create multiple views based on different data augmentations. These views are passed through a two-branch pipeline for similarity measurement (e.g., InfoNCE loss (Oord et al., 2018) or redundancy-reduction loss (Zbontar et al., 2021)). Based on this learning framework, investigations on memory queue (He et al., 2020), large batch size (Chen et al., 2020b), encoder synchronization (Chen & He, 2021), and self-distillation (Caron et al., 2021) show how to effectively learn self-supervised representations.

In a contrastive learning framework, the created views are automatically assigned with binary labels according to different image sources. The views created from the same image instance are labeled as positive, while the views from other image instances are labeled as negative. These positive

---

*Corresponding author. We provide the [homepage](#) for this project.

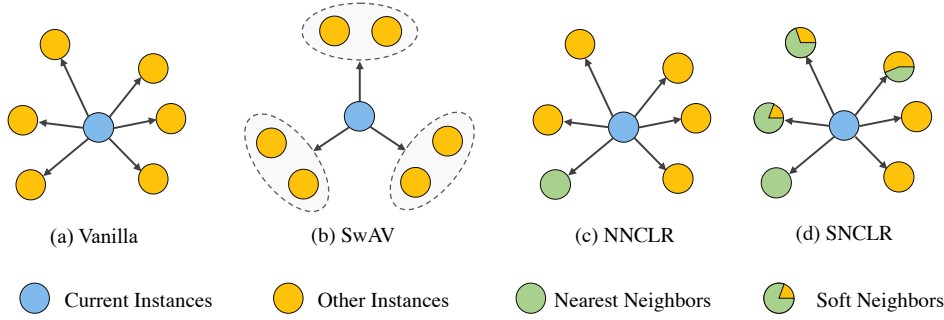

Figure 1: Training sample comparisons in the self-supervised contrastive learning framework. The vanilla contrastive learning method is shown in (a) where the current instance is regarded as positive while others are negative. In SwAV (Caron et al., 2020), the current instance is assigned to online maintained clusters for contrastive measurement. In NNCLR (Dwibedi et al., 2021), the nearest neighbor is selected as a positive instance to support the current one for the contrastive loss computation (Chen et al., 2020b). Different from these methods, our SNCLR shown in (d) measures correlations between the current instance and other instances identified from the candidate neighbor set. We define the instances that are highly-correlated to the current instance as soft neighbors (e.g., *different* pandas to the input image belonging to the 'panda' category in Fig. 3). The similarity/correlation, which is based on the cross-attention computation from a shallow attention module, denotes the instance positiveness to compute the contrastive loss. (Best viewed in colors)

and negative pairs constitute the contrastive loss computation process. In practice, we observe that this rudimentary labeling is not sufficient to represent instance correlations (i.e., similar to the semantic relations in supervised learning) between different images and may hurt the performance of learned representations. For example, when we use ImageNet data (Russakovsky et al., 2015) for self-supervised pre-training, views from the current image instance $x_0$ belonging to the 'bloodhound' category is labeled as positive, while views from other image instances belonging to the same category will be labeled as negative. Since we do not know the original training labels during SSL, images belonging to the same semantic category will be labeled differently to limit feature representations. On the other hand, for the current image instance $x_0$, although views from one image instance $x_1$ belonging to the 'walker hound' category and views from another image instance $x_2$ belonging to the 'peacock' category are both labeled as negative, the views from $x_1$ are more correlated to the views from $x_0$ than those views from $x_2$. This correlation is not properly encoded during contrastive learning as views from both $x_1$ and $x_2$ are labeled the same. Without original training labels, contrastive learning methods are not effective in capturing the correlations between image instances. The learned feature representations are limited to describing correlated visual contents across different images.

Exploring instance correlations have arisen recently in SwAV (Caron et al., 2020) and NNCLR (Dwibedi et al., 2021). Fig. 1 shows how they improve training sample comparisons upon the vanilla CLR. In (a), the vanilla CLR method labels the view from the current image instance as positive while views from other image instances as negative. This CLR framework is improved in (b) where SwAV assigns view embeddings of the current image instance to neighboring clusters. The contrastive loss computation is based on these clusters rather than view embeddings. Another improvement upon (a) is shown in (c) where nearest neighbor (NN) view embeddings are utilized in NNCLR to support the current views to compute the contrastive loss. The clustered CLR and NN selection inspire us to take advantage of both. We aim to accurately identify the neighbors that are highly correlated to the current image instance, which is inspired by the NN selection. Meanwhile, we expect to produce a soft measurement of the correlation extent, which is motivated by the clustered CLR. In the end, the identified neighbors with adaptive weights support the current sample for the contrastive loss computation.

In this paper, we propose to explore soft neighbors during contrastive learning (SNCLR). Our framework consists of two encoders, two projectors, and one predictor, which are commonly adopted in CLR framework design (Grill et al., 2020; Chen et al., 2021). Moreover, we introduce a candidate neighbor set to store nearest neighbors and an attention module to compute cross-attention scores. For the current view instance, the candidate neighbor set contains instance features from other images, but their feature representations are similar to that of the current view instance. We then

compute a cross-attention score of each instance feature from this neighbor set with respect to the current view instance. This value is a soft measurement of the positiveness of each candidate neighbor contributing to the current instance during contrastive loss computations. Basically, a higher positiveness often stands for a higher correlation between two samples. To this end, our SNCLR incorporates soft positive neighbors to support the current view instance for contrastive loss computations. We evaluate our SNCLR on benchmark datasets by using both CNN and ViT encoders. The results of various downstream recognition scenarios indicate that SNCLR improves feature representations to state-of-the-art performance.

## 2 RELATED WORKS

Our SNCLR framework explores nearest neighbors in self-supervised visual pre-training. In this section, we perform a literature review on self-supervised visual representation learning and nearest neighbor exploration in computer vision.

### 2.1 SELF-SUPERVISED VISUAL REPRESENTATION LEARNING

The objective of the SSL paradigm is to construct a pre-text feature learning process without manually labeling efforts. Using unlabeled visual data (e.g., images and videos), SSL methods design various pretexts to let these data supervise themselves. The learned features via self-supervision are more generalizable to benefit downstream scenarios where supervised learning adapts these pre-text features into specific visual tasks. Existing SSL methods can be categorized into generative and discriminative frameworks. The generative methods typically introduce an autoencoder structure for image reconstruction (Vincent et al., 2008; Rezende et al., 2014; Chen et al., 2020a), or model the data and representation in a joint embedding space (Donahue & Simonyan, 2019). Recently, breakthrough frameworks have been developed based on masked image modeling (MIM) (He et al., 2022; Bao et al., 2021; Tong et al., 2022). The visual data is partially masked and sent to an encoder-decoder structure for data recovery. By reconstructing pixels, MIM methods produce visual features that are more generalizable. However, these methods are specifically designed for vision transformers and are computationally intensive. Without downstream finetuning, off-the-shelf adoption of their pre-trained features does not perform favorably on recognition scenarios.

Apart from the generative framework, the discriminative SSL methods aim to minimize the feature distances of views augmented from the same images while maximizing views from two different images. Among discriminative methods, one of the representative arts is contrastive learning. Based on the instance discrimination paradigm, studies improve the learned representations by exploring memory bank (He et al., 2020; Pan et al., 2021), features coding (Henaff, 2020; Tsai et al., 2021), multi-modality coding (Tian et al., 2020a; Tsai et al., 2020), strong data-augmentation (Grill et al., 2020; Caron et al., 2020; Tian et al., 2020b), parameters-updating strategy (Chen & He, 2021; Grill et al., 2020) and loss redevelopment (Zbontar et al., 2021; Ge et al., 2021; Bardes et al., 2021). Among these, the neighboring-based methods (Caron et al., 2020; Dwibedi et al., 2021; Azabou et al., 2021) leverage either the cluster sets or the nearest neighbors to model the correlation between views. This design mitigates the limitation in CLR that visual data belonging to the same category will be labeled differently. In this work, we explore soft neighbors to support the current instance sample. These neighboring instances are from other images and make partial and adaptive contributions that are measured via cross-attention values.

### 2.2 NEAREST NEIGHBOR EXPLORATION IN VISUAL RECOGNITION

The nearest neighbor (NN) has been widely applied in the computer vision tasks, such as image classification (Boiman et al., 2008; McCann & Lowe, 2012), object detection/segmentation (Harini & Chandrasekar, 2012), domain adaption (Yang et al., 2021; Tang et al., 2021; Tommasi & Caputo, 2013), vision alignment (Thewlis et al., 2019; Dwibedi et al., 2019), and few-shot learning (Wang et al., 2019). Generally, the nearest neighbor method explores the comprehensive relations among samples to facilitate the computer vision tasks. Earlier attempts have also been made in the field of self-supervised visual pre-training. For example, in the contrastive learning framework, different from comparing representations directly, SwAV (Caron et al., 2020) additionally sets up a prototype feature cluster and simultaneously maintains the consistency between the sample features and the

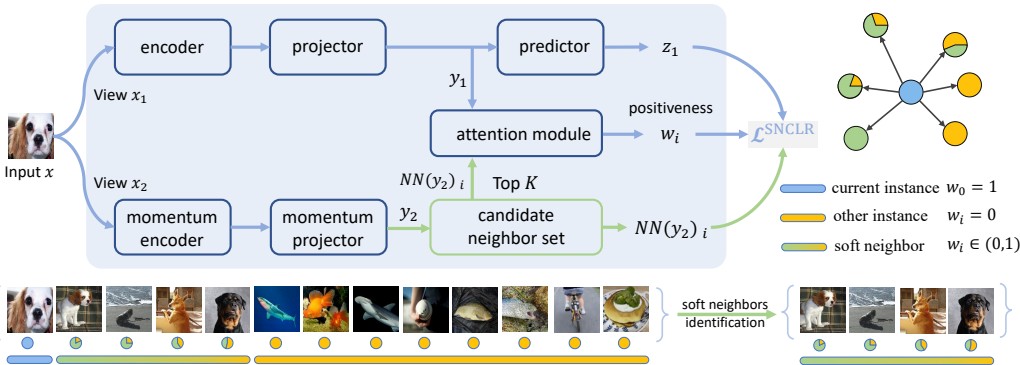

Figure 2: An overview of SNCLR. Given an input instance $x$, we obtain two views (i.e., $x_1$ and $x_2$). After projection, we obtain $y_1$ and $y_2$. We use $y_2$ to select $K$ candidates from the candidate neighbor set $\mathcal{C}$ where samples are from other images. For each selected sample $\mathrm{NN}(y_2)_i$ ($i \in [1, ..., K]$), we sent it to the attention module together with $y_1$ to compute a cross-attention value $w_i$. This value is the soft measurement of $\mathrm{NN}(y_2)_i$ for contrastive computation.

prototype features. More recently, NNCLR (Dwibedi et al., 2021) utilizes an explicit support set (also known as a memory queue/bank) for the purpose of nearest neighbor mining. In these methods, the neighbors are regarded as either 0 or 1 to contribute to the current instance sample. This is inaccurate as we observe that neighbors are usually partially related to the current sample. Different from the above methods, we leverage an attention module to measure the correlations between neighboring samples and the current one. The instance correlations are formulated as positiveness scores during contrastive loss computations.

## 3 PROPOSED METHOD

Fig. 2 shows an overview of our SNCLR. We construct a candidate neighbor set to support the current view during contrastive learning. The support extent is measured softly as positiveness via cross-attention computations. We first revisit the CLR framework, then illustrate our proposed SNCLR and its training procedure. Moreover, we visualize selected neighbors for an intuitive view of how they positively support the current instance sample.

### 3.1 REVISITING CONTRASTIVE LEARNING

We show how contrastive learning works based on our framework in Fig. 2. Given an input instance sample $x$ from a batch $\chi$, we create two augmented views $x_1$ and $x_2$. After feature extraction and projection, we obtain their corresponding outputs $y_1$ and $y_2$. For other $N-1$ instances from this batch, we denote their projected view representations as $y_2^{i-}$ where $i \in [1, 2, ..., N-1]$. The $y_2^{i-}$ is not shown in the figure for simplicity. The contrastive loss can be generally written as:

$$\mathcal{L}_x = -\log \frac{\exp\left(\mathrm{sim}(y_1, y_2)/\tau\right)}{\exp\left(\mathrm{sim}(y_1, y_2)/\tau\right) + \sum_{i=1}^{N-1} \exp\left(\mathrm{sim}(y_1, y_2^{i-})/\tau\right)}, \tag{1}$$

where $x \in \chi$, $\tau$ is a temperature parameter, $y_1$ and $y_2$ are positive pairs, $y_1$ and $y_2^{i-}$ are negative pairs. The $\mathrm{sim}(\cdot)$ is the operator for similarity measurement, which is usually set as cosine distance (Chen et al., 2020b; He et al., 2020). We can also use only positive samples and substitute $y_1$ with $z_1$ for sample similarity computation (Grill et al., 2020; Chen & He, 2021), where we use $z_1$ to denote the projected representation of $y_1$ via a predictor as shown in Fig. 2 (e.g., the sequentially-connected MLP layers). After computing the loss, momentum update (He et al., 2020) or partial gradient stop (Chen & He, 2021) is utilized to train encoders. Note that in Eq. 1 all the instances other than the current one are labeled as negative. To explore their correlations, we introduce our SNCLR as follows.

### 3.2 SOFT NEIGHBORS CONTRASTIVE LEARNING (SNCLR)

Our SNCLR introduces the adaptive weight/positiveness measurement based on a candidate neighbor set $\mathcal{C}$. We follow the CLR framework illustrated in Sec. 3.1. For the current instance $x$, we cultivate

a candidate neighbor set, where we select $K$ candidate neighbors with projected representations $\mathrm{NN}(y_2)_i$ ($i \in [1, 2, ..., K]$). The $\mathrm{NN}(\cdot)$ denotes the nearest neighbor identification operation in the projected feature space. For each identified neighbor $\mathrm{NN}(y_2)_i$, we sent it to the attention module together with $y_1$ for cross-attention computations. This attention module predicts a positiveness value $w_i$. We use this value to adjust the contributions of $\mathrm{NN}(y_2)_i$ to $z_1$ in contrastive learning. The loss function of our SNCLR can be written as:

$$\mathcal{L}_x^{\mathrm{SNCLR}} = -\frac{1}{N} \log \frac{\sum_{i=0}^{K} w_i \cdot \exp\left(z_1 \cdot \mathrm{NN}(y_2)_i / \tau\right)}{\sum_{i=0}^{K} \exp\left(z_1 \cdot \mathrm{NN}(y_2)_i / \tau\right) + \sum_{i=1}^{N-1} \sum_{j=0}^{K} \exp\left(z_1 \cdot \mathrm{NN}(y_2^{i-})_j / \tau\right)}, \quad (2)$$

where $z_1$ is the predictor output, we use $\mathrm{NN}(y_2)_0$ to denote $y_2$, use $\mathrm{NN}(y_2^{i-})_0$ to denote $y_2^{i-}$, $w_0$ is set as 1, $\mathrm{NN}(y_2^{i-})$ is the nearest neighbors search of each $y_2^{i-}$ to obtain $K$ neighbors (i.e., $\mathrm{NN}(y_2^{i-})_j$, $j \in [1, 2, ..., K]$). The $z_1$ and $\mathrm{NN}(y_2)_i$ are partially positive pairs, while the $z_1$ and $\mathrm{NN}(y_2^{i-})_j$ are negative pairs. If we exclude the searched neighbors of $y_2$ and $y_2^{i-}$, Eq. 2 will degenerate to Eq. 1.

**Candidate Neighbors.** We follow Dwibedi et al. (2021) use a queue as candidate neighbor set $\mathcal{C}$, where each element is the projected feature representation from the momentum branch. When using sample batches for network training, we store their projected features in $\mathcal{C}$ and adopt a first-in-first-out strategy to update the candidate set. The length of this set is relatively large to sufficiently provide potential neighbors. When identifying $K$ candidate neighbors based on this set, we use the cosine similarity as the selection metric. The nearest neighbors search can be written as follows:

$$\mathrm{NN}(y_2) = \underset{c \in \mathcal{C}}{\arg\max} \ (\cos(y_2, c), \mathrm{top_n} = K), \quad (3)$$

where both $y_2$ and $c \in \mathcal{C}$ are normalized before computation. The $\mathrm{NN}(\cdot)$ operation is utilized for both $y_2$ and $y_2^{i-}$ to identify the top-$K$ corresponding neighbors.

**Positiveness Predictions.** We quantitatively measure the correlations of the selected $K$ neighbors with respect to the current instance sample. Fig. 2 shows an intuitive view of the candidate neighbors. We observe that the first two neighbors belong to the same category as the input sample (i.e., 'Cavalier King Charles Spaniel'), while the third and fourth neighbors are partially related (i.e., these two samples belong to the 'dog' category but not in 'Cavalier King Charles Spaniel' breed). Besides, other samples do not belong to dogs. To this end, we introduce an attention module to softly measure (not in a binary form) their correlations (also known as positiveness in this paper). This module contains two feature projection layers, a cross-attention operator, and a nonlinear activation layer. Given two inputs $y_1$ and $\mathrm{NN}(y_2)_i$, the positiveness score can be computed as:

$$w_i = \frac{1}{\gamma_i} \mathrm{Softmax}\left[f_1(y_1) \times f_2(\mathrm{NN}(y_2)_i)^\top\right], \quad (4)$$

where $f_1(\cdot)$ and $f_2(\cdot)$ are the projection layers of this attention module, $\gamma_i$ is scaling factor to adjust positiveness $w_i$. We have empirically tried using different structures of this attention module, including using parametric linear projections for $f_1(\cdot)$ and $f_2(\cdot)$, and adding more projection layers after the nonlinear activation layer. The results of these trials are inferior than simply using Eq. 4 by setting $f_1(\cdot)$ and $f_2(\cdot)$ as identity mappings without parameters. This may be due to the self-supervised learning paradigm, where the objective is to effectively train the encoder. A simplification of this attention module (i.e., with less attentiveness) will activate the encoder to produce more attentive features during pretext training.

**Analysis: Computing contrastive loss with more instances.** The soft weight cross-attention design empowers us to properly model the correlations between the current instance and others. This design enables us to leverage more instances to compute the contrastive loss and thus brings two advantages. *First,* we enrich the diversity of positive instances to support the current one in a soft and adaptive way. Different from the original SSL frameworks that align augmented views with the current instance, SNCLR explores correlations between different instances. At the same time, our SNCLR mitigates the limitation that instances belonging to the same semantic category are labeled oppositely in CLR. *Second,* SNCLR introduces more negative samples for discrimination (i.e., the yellow instances in Fig. 2). The number of negative samples introduced in SNCLR is $N$ times larger than existing methods (Chen et al., 2021; Dwibedi et al., 2021), where negative samples are from the current mini-batch. Using more negative samples and partially related positive samples benefits SNCLR in discriminating uncorrelated instances while absorbing correlated instances during pretext training.

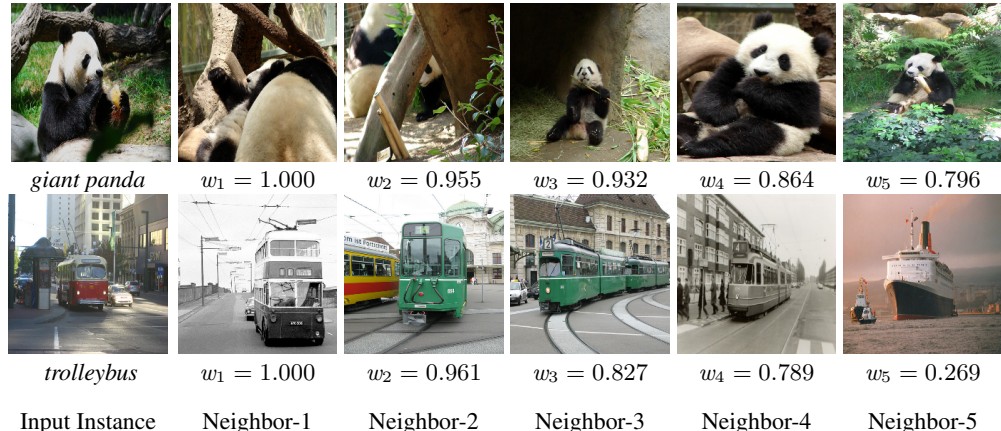

| | | | | | |
|---|---|---|---|---|---|
| *giant panda* | $w_1 = 1.000$ | $w_2 = 0.955$ | $w_3 = 0.932$ | $w_4 = 0.864$ | $w_5 = 0.796$ |
| *trolleybus* | $w_1 = 1.000$ | $w_2 = 0.961$ | $w_3 = 0.827$ | $w_4 = 0.789$ | $w_5 = 0.269$ |
| Input Instance | Neighbor-1 | Neighbor-2 | Neighbor-3 | Neighbor-4 | Neighbor-5 |

Figure 3: Neighbors display. We show input instances in the first column. Their nearest 5 neighbors are shown on the corresponding row for clarification. We normalize the scores to clarify its positive contribution to the current sample.

**Network Training.** We compute contrastive loss via Eq. 2, and update network parameters of the upper branch shown in Fig. 2 via back-propagation. Afterwards, we follow (Grill et al., 2020; Chen et al., 2021) to perform a moving average update on the momentum networks (i.e., the momentum encoder and the momentum projector). We refer the readers to Appendix A for more details on the network training details.

**Neighbors Display.** We show selected neighbors with var input instances in Fig. 3. The encoder backbone of ViT-S is utilized for visualization, with 300 epochs of training. Given an input instance, we use Eq. 3 to select top-$K$ neighbors and show them in this figure. We use an input instance from the 'giant panda' category in the first row. The category is unknown during SNCLR. The top 5 neighbors all belong to the 'giant panda' with high positiveness values. This indicates our SNCLR effectively selects highly-correlated neighbors and computes high positiveness scores for contrastive loss computations. On the second row, the input instance is from the 'trolleybus' category. The top 4 neighbors are from the same category with high positiveness scores, while the 5-th neighbor is from the 'ocean liner' category with a smller positiveness score. That's to say, the high score of $w_i$ corresponds to the same category while the lower score corresponds to less-correlated contents, which shows that our attention module is effective in modeling correlations between the input instance and each candidate neighbor. As shown in Eq. 4, we accordingly scale the positiveness with a factor. More visualization examples are presented in Appendix C.1.

## 4 EXPERIMENTS

In this section, we perform experimental validation on our SNCLR framework. The implementation details will be first introduced. Then, we compare our proposed SNCLR to state-of-the-art SSL methods on prevalent vision tasks, including image classification, object detection, and instance segmentation. In the end, the critical component will be ablated for effectiveness validation.

### 4.1 IMPLEMENTATION DETAILS

**Network architectures.** We adopt prevalent vision backbones (i.e., Vision Transformer (ViT) (Dosovitskiy et al., 2021) and ResNet (He et al., 2016)) as the visual encoders. To fully explore the potential of our proposed SNCLR, we adopt three different encoder architectures for validation, including ResNet-50, ViT-S, and ViT-B. For the ResNet, the features by the final block are further processed by a global context pooling operator (Gholamalinezhad & Khosravi, 2020) to extract the representations. For the ViTs, we follow (Dosovitskiy et al., 2021; Chen et al., 2021; Caron et al., 2021) to leverage the `[CLS]` tokens as the representation vectors. The designs of the projectors and the predictors follow those from (Grill et al., 2020), where there are two fully-connected layers with a batch normalization (Ioffe & Szegedy, 2015) and a ReLU activation (Agarap, 2018) in between.

Table 1: Linear evaluations on ImageNet with top-1 accuracy (in %). Both the ResNet and ViT encoders are utilized. Prevalent methods, including SimLCR (Chen et al., 2020b), MoCo (He et al., 2020; Chen et al., 2020c; 2021), SimSiam (Chen & He, 2021),InfoMin (Tian et al., 2020b),SwAV (Caron et al., 2020), BYOL (Grill et al., 2020), NNCLR (Dwibedi et al., 2021), DINO (Caron et al., 2021), MAE (He et al., 2022), and CAE (Chen et al., 2022). We highlight best results under the same model parameters constraints in **bold.** ∗ denotes the experimental results implemented by ourselves since there are no direct results provided. For ResNet, we show the accuracy results of the encoder trained with 100, 200, 400 and 800 epochs. For ViTs, the performances are reported with the encoders trained with 100, 200 and 300 epochs. All the results are reported without the multi-crop augmentation.

(a) ResNet-50.

| Method | 100 | 200 | 400 | 800 |
|---|---|---|---|---|
| SimCLR | 66.5 | 68.3 | 69.8 | 70.4 |
| InfoMin Aug | - | 70.1 | - | - |
| SwAV | 66.5 | 69.1 | 70.7 | 71.8 |
| MoCo v2 | 67.4 | 69.9 | 71.0 | 72.2 |
| MoCo v3 | 68.1 | 72.2 | 73.0 | 74.1 |
| SimSiam | 68.1 | 70.0 | 70.8 | 71.3 |
| NNCLR | 69.2 | 70.7 | 74.0 | 74.5 |
| BYOL | 66.5 | 70.6 | 73.2 | 74.3 |
| Barlow Twins | - | - | 72.5 | 73.2 |
| SNCLR(ours) | **69.6** | **72.4** | **74.5** | **75.3** |

(b) ViT-S.

| Method | 100 | 200 | 300 |
|---|---|---|---|
| BYOL | - | - | 71.0 |
| MoCo v2 | - | - | 71.6 |
| MoCo v3∗ | 67.5 | 71.8 | 72.5 |
| SwAV | - | - | 67.1 |
| DINO | - | - | 72.5 |
| NNCLR∗ | 66.8 | 70.9 | 71.5 |
| SimCLR | - | - | 69.0 |
| CAE | - | - | 51.8 |
| SNCLR(ours) | **68.5** | **72.5** | **73.4** |

(c) ViT-B.

| Method | 100 | 200 | 300 |
|---|---|---|---|
| BYOL | - | - | 73.9 |
| MoCo v3 | - | - | 76.2 |
| NNCLR∗ | 69.1 | 71.9 | 75.9 |
| SimCLR | - | - | 73.9 |
| MAE | - | - | 61.5 |
| MAE_{ViT-L} | 57.3 | 64.4 | 67.3 |
| CAE | - | - | 64.1 |
| SNCLR(ours) | **71.4** | **74.3** | **76.8** |

**Training configurations.** The pretext training is conducted on the ImageNet-1k dataset (Russakovsky et al., 2015) without labels. We follow the data augmentation strategies from (Grill et al., 2020) to create distorted views, which include the resolution adjustment, random horizontal flip, color distortions, Gaussian blur operations, etc. More details about data augmentation are described in Appendix A. For ResNet encoders, we adopt the LARS optimizer (You et al., 2017) with cosine annealing schedule (Loshchilov & Hutter, 2016) to train the networks for 800 epochs with a warm-up of 10 epochs. The learning rate is scaled linearly with respect to the batch size via $lr = 0.3 \times BatchSize/256$ (Goyal et al., 2017), where 0.3 denotes the base learning rate. The update coefficient of the momentum network branches is set as 0.99. Unless otherwise stated, we utilize 30 soft neighbors to train SNCLR. Different from ResNet, we leverage AdamW optimizer (Loshchilov & Hutter, 2017) to train ViTs for 300 epochs with a warm-up of 40 epochs. Following (Chen et al., 2021), the base learning rate is configured as $1.5 \times 10^{-4}$. We refer the readers to Appendix A for more detailed training configurations.

## 4.2 COMPARISON TO STATE-OF-THE-ART APPROACHES

The representations learned by SNCLR are compared with other state-of-the-art frameworks in various downstream tasks, including image classification, object detection, and instance segmentation.

**Linear evaluation on image classification.** Following the standard linear classification protocol (Grill et al., 2020), we freeze the parameters of the encoder backbones and additionally train a linear head. We use the SGD optimizer, with a momentum of 0.9, a batch size of 4096, and a weight decay of 0, during linear training. The base learning rate is set as 0.1 for ResNets and 3 for ViTs.

As shown in Table 1, we perform linear probe on encoders (i.e., fine-tuning the linear head with 90 epochs while keeping the encoders fixed) and report the linear evaluation with top-1 accuracy. Table 1a presents the classification results of ResNet-50 encoders with various self-supervised frameworks. It validates that our SNCLR consistently outperforms the prevalent SSL methods at different training epochs. For example, SNCLR obtains a 74.5% top-1 accuracy with only 400 training epochs, which is even higher than almost all SSL frameworks trained with 800 epochs (e.g., 74.3% with BYOL (Grill et al., 2020), and 71.8% with SwAV (Caron et al., 2020)). Even compared to the methods (Dwibedi et al., 2021) explicitly involving the neighbor information, SNCLR achieves the best performance of 75.3% top-1 accuracy at 800 epoch. We also evaluate the effectiveness of SNCLR on ViT encoders. Quantitative results in Table 1b reveal that absorbing the highly-correlated neighbors empowers the various ViT encoders in extracting more discriminating representations as well, thus facilitating image classifications (e.g., 73.4% with ViT-S and 76.8% with ViT-B).

Table 2: Image classification by using semi-supervised training on ImageNet. $^*$ denotes the results implemented by ourselves since there are no direct results provided.

(a) Classification accuracy via ResNet-50.

| Method | Epoch | Top-1 1% | Top-1 10% | Top-5 1% | Top-5 10% |
|---|---|---|---|---|---|
| Supervised | - | 25.4 | 56.4 | 48.4 | 80.4 |
| SimCLR | 800 | 48.3 | 65.6 | 75.5 | 87.8 |
| NNCLR$^*$ | 400 | 55.1 | 69.0 | 79.2 | 88.6 |
| MoCo v3$^*$ | 400 | 51.2 | 62.2 | 76.1 | 82.4 |
| BYOL | 800 | 53.2 | 68.8 | 78.4 | 89.0 |
| SNCLR(Ours) | 400 | **56.0** | **69.2** | **80.1** | **89.1** |

(b) Classification accuracy via ViT-S.

| Method | Epoch | Top-1 1% | Top-1 10% | Top-5 1% | Top-5 10% |
|---|---|---|---|---|---|
| MoCo v3$^*$ | 300 | 46.9 | 58.3 | 72.1 | 79.4 |
| BYOL$^*$ | 300 | 44.2 | 54.5 | 71.8 | 77.9 |
| NNCLR$^*$ | 300 | 45.8 | 55.6 | 71.7 | 78.0 |
| DINO$^*$ | 300 | 50.9 | 61.4 | 75.3 | 87.9 |
| SNCLR(ours) | 300 | **51.2** | **61.9** | **76.0** | **88.4** |

**Linear evaluation on image classification with semi-supervised learning.** The representations are also evaluated in the semi-supervised setting. After pretext training, we add an extra classifier head and fine-tune the overall networks with a certain subset of ImageNet training samples. Following the previous works (Zbontar et al., 2021; Grill et al., 2020), we leverage 1% and 10% of the training set for fine-tuning. As shown in Table 2, by fine-tuning the networks on 10% training dataset from ImageNet-1k, SNCLR outperforms other methods with both the RersNet and ViT backbones. The considerable top-1 accuracy in Table 2 validates that encoders trained with SNCLR are able to produce more generalizable representations.

**Evaluation on object detection and segmentation.** To compare the transfer learning abilities of the learned representations, we conduct evaluations on object detection and instance segmentation. We adopt the prevalent COCO datasets (Lin et al., 2014). The pre-trained encoder of SNCLR is extracted and integrated into Mask R-CNN (He et al., 2017). For fair comparisons, the same encoder (i.e., ResNet-50) is leveraged as the visual backbones. We follow the standard training configurations provided in (Chen et al., 2019) to fine-tune the detectors with a '1x' schedule (90k iterations).

Table 3 shows the evaluation results. Compared to the supervised methods in the first block, SNCLR advances both object detection (by 1.0% $\mathrm{AP}^{bb}$) and instance segmentation (by 1.0% $\mathrm{AP}^{mk}$). Besides, SNCLR outperforms favorably prevalent SSL methods. Taking the visual encoder trained with 200 epochs for example, SNCLR achieves the comparable performance trained with more epochs (e.g., 39.4% $\mathrm{AP}^{bb}$ for BYOL with 800 epochs) or better performance (e.g., +2.5% $\mathrm{AP}^{bb}$ for Barlow Twins with 1000 epochs). Similar results could be observed in instance segmentation scenarios as well, where SNCLR consistently outperforms other SSL methods. The performance profits indicate that encoding the contents correlation facilitates exploring rich information in samples, thus benefiting the visual encoders in generating more transferable representations for detection and segmentation tasks.

### 4.3 ABLATION STUDIES

In the SNCLR framework, correlation extent among the current instance and neighbors in the candidate set are encoded as positiveness for pretext training. Thus, we investigate the influence of the main components, including the positiveness, selected neighbors, and candidate neighbor set. Unless otherwise stated, the ViT-S encoder pre-trained with 300 epochs is utilized for ablation studies.

**Importance of positiveness.** Our main contribution is encoding the correlations as positiveness to train the visual encoders. We analyze the importance of positiveness to the recognition performance in the second block of Table 4. Taking 30 neighbors for example, when we remove the positiveness and label all the neighbors as exactly positive samples, the top-1 accuracy decreases from 73.38% to 72.24%. It indicates that the binary labeling strategy is insufficient to measure the correlation extent at the feature space, thus causing the encoders to produce less discriminating representations compared with soft labeling methods.

**Importance of neighbors.** We analyze how the selected neighbors affect the recognition performance as well. As shown in the first two blocks of Table 4, the visual encoders achieve a performance gain of +1.2% (i.e., 72.24% *v.s.* 71.17%) with the assistance of neighbors during the pretext training. The performance profits may contribute to the abundant comparisons/supports from positive samples, which facilitates the current instance to align more highly-correlated views. In the third block of

Table 3: Transfer learning to object detection and instance segmentation with the Mask R-CNN R50-FPN detector. We highlight the best two results in **bold**. We compare SNCLR with prevalant methods, including PIRL (Misra & Maaten, 2020), Barlow Twins (Zbontar et al., 2021), DetCo (Xie et al., 2021) and etc. Our method achieves favorable detection and segmentation performance.

| Method | Epoch | COCO det. | | | COCO instance seg. | | |
|---|---|---|---|---|---|---|---|
| | | $AP^{bb}$ | $AP^{bb}_{50}$ | $AP^{bb}_{75}$ | $AP^{mk}$ | $AP^{mk}_{50}$ | $AP^{mk}_{75}$ |
| Rand Init | - | 31.0 | 49.5 | 33.2 | 28.5 | 46.8 | 30.4 |
| Supervised | 90 | 38.9 | 59.6 | 42.7 | 35.4 | 56.5 | 38.1 |
| PIRL | 200 | 37.5 | 57.6 | 41.0 | 34.0 | 54.6 | 36.2 |
| SwAV | 200 | 38.5 | **60.4** | 41.4 | 35.4 | 57.0 | 37.7 |
| MoCo | 200 | 38.5 | 58.9 | 42.0 | 35.1 | 55.9 | 37.7 |
| MoCo v2 | 200 | 38.9 | 59.4 | 42.4 | 35.5 | 56.5 | 38.1 |
| MoCo v2 | 800 | 39.4 | 59.9 | **43.0** | 35.8 | 56.9 | 38.4 |
| Barlow Twins | 1000 | 36.9 | 58.5 | 39.7 | 34.3 | 55.4 | 36.5 |
| BYOL | 200 | 39.1 | 59.5 | 42.7 | 35.6 | 56.5 | 38.2 |
| BYOL | 400 | 39.2 | 59.6 | 42.9 | 35.6 | 56.7 | 38.2 |
| BYOL | 800 | **39.4** | 59.9 | **43.0** | 35.8 | 56.8 | **38.5** |
| DetCo | 200 | **39.4** | 59.2 | 42.3 | 34.4 | 55.7 | 36.6 |
| SNCLR (ours) | 200 | **39.4** | 60.3 | **43.0** | **36.0** | **57.1** | 38.3 |
| SNCLR (ours) | 800 | **39.9** | **60.5** | **43.7** | **36.4** | **57.3** | **39.0** |

Table 4: Analysis on the positiveness and number of neighbors $K$.  Table 5: Analysis on set length.

| neighbor | positiveness | $K$ | Top-1 |
|---|---|---|---|
| ✗ | ✗ | - | 71.17 |
| ✓ | ✗ | 30 | 72.24 (+1.07) |
| ✓ | ✓ | 30 | **73.38** (+2.21) |
| ✓ | ✓ | 10 | 72.81 (+1.64) |
| ✓ | ✓ | 30 | **73.38** (+2.21) |
| ✓ | ✓ | 50 | 72.92 (+1.75) |

| length | Top-1 |
|---|---|
| 0 | 71.17 |
| 8000 | 72.25 (+1.08) |
| 16000 | 72.67 (+1.50) |
| 32000 | 72.43 (+1.26) |
| 64000 | 73.02 (+1.85) |
| 128000 | **73.38** (+2.21) |

Table 4, we also conduct experiments to analyze the influence of the number of neighbors. It's empirically found that there is a trade-off for the neighbor number. On the one hand, sufficient neighbors are able to enrich the diversities of the positive comparisons. On the other hand, there is a certain probability of including some correlation-agnostic instances, which shall be the negative samples, into positive neighbors. Involving the less-correlated as positive samples may disturb the training process and hurts the recognition performance. Our results show that 30 neighbors achieve the best accuracy. Thus, we adopt 30 neighbors in our experiments for simplicity.

**Analysis on the neighbor set length.** We empirically found that a large set increases the recognition performance. For example, results in Table 5 show that the encoder trained with the largest set achieves the best accuracy of 73.38%. It indicates that with the capacity of the set increasing, it will be possible for the current instance to find more correlated neighbors to softly support itself.

## 5 CONCLUSIONS

In this work, we propose a contrastive learning framework, termed SNCLR, to encode the correlation extent between instances as positiveness for softly supporting the positive samples. Contrary to the previous works that utilize binary weights to measure the positive and negative pairs, our SNCLR leverages the soft weights to adaptively support the current instances from their neighbors. Thanks to sufficiently exploring the correlation among samples, our proposed SNCLR facilitate both the ResNet and ViT encoders in producing more generalizable and discriminating representations. Experiments on various visual tasks, including image classification, object detection, and instance segmentation, indicate the effectiveness of our proposed SNCLR.

**Acknowledgement.** This work is supported by the General Research Fund of HK No.27208720, No.17212120, and No.17200622.

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

In this appendix material, the experimental details will be further described, including the network architectures, data augmentations, and basic training configurations. In the end, more visualization results on the selected neighbors and experimental analysis will be presented.

## A    EXPERIMENTAL DETAILS

**Network Architectures.**  We adopt both the ResNet-50, ViT-S, and ViT-B as visual encoders for experimental comparison.  Specifically, as for ResNet-50, we follow the implementation of `Torchvision Toolkit` (Paszke et al., 2019). The 2-D feature map produced by the ResNet-50 encoder is globally pooled to produce the 1-D representation. For ViTs, we leverage the implementation from `timm toolkit` (Wightman, 2019) and `DeiT` (Touvron et al., 2021). The patch resolution of $16 \times 16$ is adopted for both ViT-S and ViT-B. The attached `[CLS]` token is extracted as the produced representations. Each projector and predictor in our framework consists of a fully-connected layer with a batch normalization (Ioffe & Szegedy, 2015) and a ReLU activation layer (Agarap, 2018), followed by another fully-connected layer. Note that we utilize the identity mapping for the vector projection in the cross-attention module. Thus, there are no network parameters in this block.

**Image Processing.** We here illustrate how we create two distorted views via data augmentation. For the pretext training stage, we follow (Grill et al., 2020) to conduct two groups of data augmentation on the input image.  Specifically, we first randomly crop local patches from the inputs and resize them to $224 \times 224$ resolution. Then the random color distortions, the solarization, and the Gaussian blur operations are applied to these patches. In the end, we perform the random horizontal flip and normalization on the processed samples to feed the networks for training. For the data augmentation in the linear evaluation procedure, we follow (Chen et al., 2021; 2020c) to include random cropping, resizing, and horizontal flip for simple data transformation.

**Training Configurations.** The training configurations are different among ResNets and ViTs. For ResNet-50, the LARS optimizer (You et al., 2017) is adopted to train the networks for a total of 800 epochs. The learning rate follows a cosine annealing schedule without restarting (Grill et al., 2020; Loshchilov & Hutter, 2016). For the first 10 epochs, we warm-up the learning rate linearly from the initial $\text{lr} = 10^{-6}$ to the base learning rate ($\text{lr} = 0.3 \times \text{BatchSize}/256$). The network updating coefficient is fixed as 0.99. We train SNCLR using 32 NVIDIA A100 GPUs with a batch size of 4096. Unlike the above training protocols, several main components are utilized for training ViTs. We leverage the AdamW optimizer (Loshchilov & Hutter, 2017) with a weight decay of 0.1 for training. The base learning rate is configured as ($\text{lr} = 1.5 \times 10^{-4} \times \text{BatchSize}/256$) with a linear warm-up of 40 epochs. Besides, we follow the previous works (Caron et al., 2021; Chen et al., 2021) to configure the total training schedule as 300 epochs. The other configurations (e.g., batch size, cosine annealing learning rate, etc.) are identical to the ones utilized in training ResNets. During the initial training stage, we observe that neighbors are not correctly identified. This may be due to the premature feature representations from the encoder. To mitigate incorrect support neighbor selections, we degrade Eq. 2 to Eq. 1 during the first $m$ training epochs by not using neighbors. Then, we resume using Eq. 2 in SNCLR pretext training.

## B    VISUALIZATION OF NEIGHBORS

In this subsection, we present extended examples with more neighbors in Fig. 4. It's empirically found that the neighbors consistently share the related information/content with the input samples (e.g., the 'bananas' in the first group and the 'monkey' in the second group). We also observe that even though the input images consist of multiple objects (e.g., the fruits in the third group), our proposed methods are still able to index the highly-correlated instances, including related objects. The support from such strong correlated neighbors facilitates our SNCLR in the pretext training stage to produce more discriminating features.

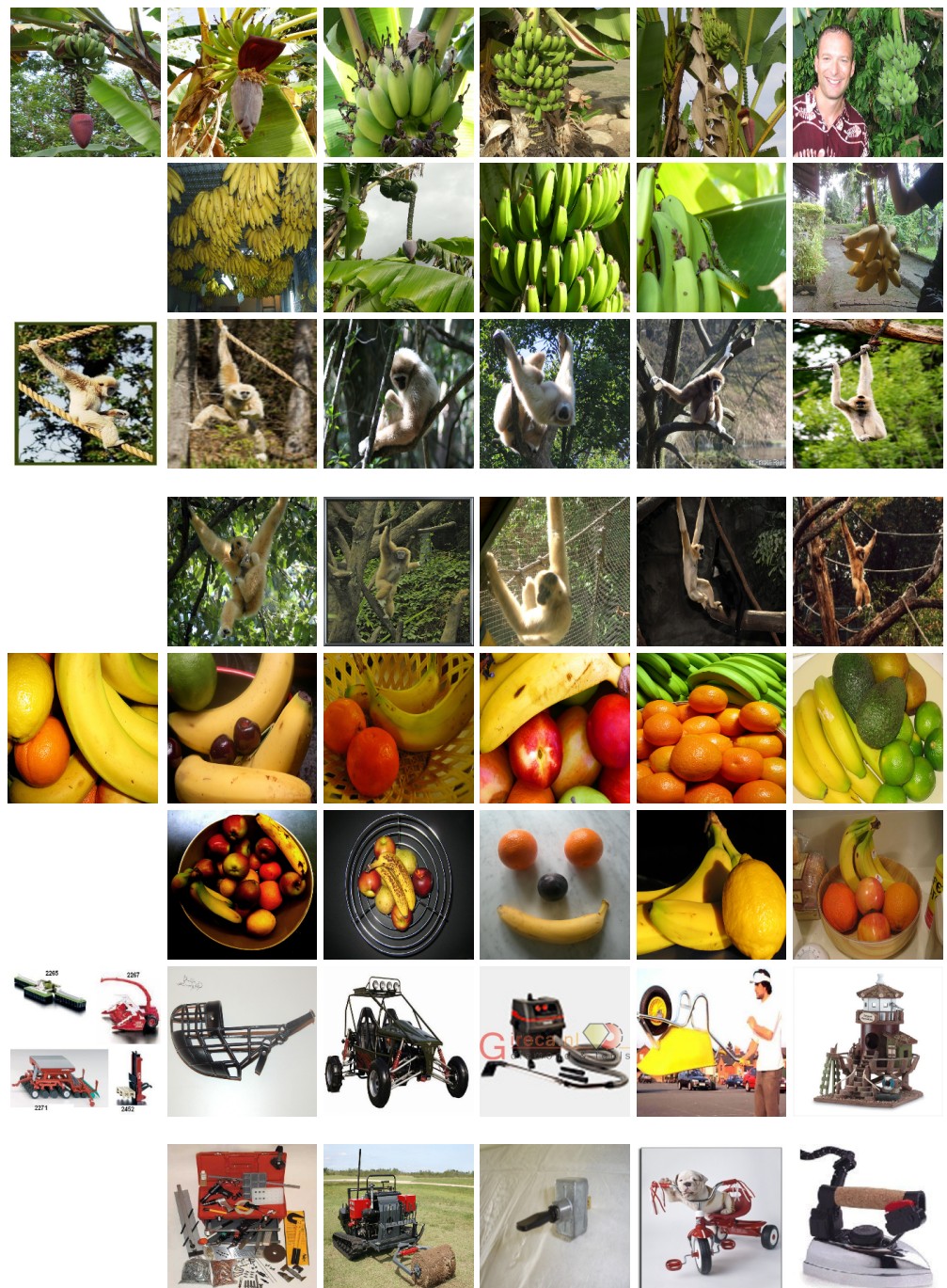

Figure 4: **Neighbors display.** We show the input instance on the first column. Unlike Fig. 3, there are 10 neighbors in total presented with the ranking from highest positiveness score to the lowest score.

Table 6: Analysis on the memory overhead and computational complexity. We adopt the ViT-Small structure for comparison. All the experiments are tested on 8 NVIDIA A100-40G GPUs.

| Set Length | 8000 | 16000 | 32000 | 64000 | 128000 |
|---|---|---|---|---|---|
| Additional GPU Memory (MB) | +8.19 | +16.38 | +32.77 | +65.54 | +131.07 |
| Seconds per batch | 2.687 | 2.717 | 2.755 | 2.824 | 2.883 |

## C ADDITIONAL EXPERIMENTAL RESULTS AND ANALYSIS

### C.1 MEMORY USAGE AND COMPUTATIONAL COMPLEXITY

Since a large pool of neighborhood set should be maintained in the proposed SNCLR, we study the empirical computation cost and memory usage with respect to the set length. The additional GPU memory and training speed are chosen as the evaluation metrics. All the experiments are tested with ViT-Small on 8 NVIDIA A100-40G GPUs. For the training speed, all the time duration is measured in seconds averaged over 20 epochs. The results in Table 6 show that the computational overhead increases marginally with the set length. Specifically, when the set contains 128000 candidates of which the dimension is 256, it additionally costs 131.07 MB of GPU memory (i.e., 30420 MB v.s. 30551MB (+0.43%)). Considering the performance gains brought by the bigger set, such computational complexity is acceptable.

### C.2 ABLATION STUDIES ON THE INDIVIDUAL BENEFIT OF NEIGHBORS

We study the individual performance of neighbors on either positive instances or negative instances. Specifically, three different strategies are adopted to train ViT-Small for 300 epochs, including 1) applying the soft neighbors on the positive instances; 2) applying the soft neighbors on the negative instances; 3) applying the neighbors on both positive and negative samples. Experiments summarized in the Table 7 show that implying the neighbors on both positive and negative samples achieves the best result in the linear probe evaluation. Besides, implying neighbors on either positive or negative samples leads an inferior performance compared with the fully-implemented SNCLR, while being better than the baseline results. It indicates that the full implementation enables to leverage more instances for discrimination and thus brings two advantages: 1) it enriches the diversity of positive instances for adaptive clustering; 2) It introduces more negative samples for sufficient discrimination.

Table 7: Analysis on the individual benefits of soft neighbors.

| Backbone | Positive | Negative | Top-1 (%) |
|---|---|---|---|
| ViT-Small | ✗ | ✗ | 71.17 |
| ViT-Small | ✓ | ✗ | 71.88 (+0.71) |
| ViT-Small | ✗ | ✓ | 72.43 (+1.26) |
| ViT-Small | ✓ | ✓ | 73.38 (+2.21) |

### C.3 ANALYSIS ON THE SNCLR

We provide a conceptual comparison between previous methods (i.e., SwAV Caron et al. (2020), NNCLR Dwibedi et al. (2021)) and ours to illustrate why SNCLR performs better. The differences between ours and existing methods are summarized as follows: 1) Unlike SwAV Caron et al. (2020) that utilizes the fixed prototype to include the semantic-correlated samples, we explicitly compute the correlation/similarity extent via cross-attention and further encoder the correlation/similarity into the training objective. To this end, the proposed SNCLR framework is able to adaptively distribute the representations according to the sample semantics during the training process. 2)Compared with NNCLR Dwibedi et al. (2021), SNCLR abandons the traditional binary instance discrimination. Instead, SNCLR first includes more positive neighbors (larger than one) and adopts soft discrimination to fully support the current instance via cross-attention similarities. To summarize, SNCLR explores an adaptive and soft way to treat the images that correlate similarly to the current instance, thus being more effective in the contrastive learning.

