# OpenReview forum: "Soft Neighbors are Positive Supporters in Contrastive Visual Representation Learning"
_ICLR.cc/2023/Conference — ICLR 2023 poster_

### Official Review · Reviewer_EMnc · 2022-10-21

**Confidence:** 4
**Correctness:** 3
**Technical Novelty And Significance:** 3
**Empirical Novelty And Significance:** 3
**Recommendation:** 8

**Clarity, Quality, Novelty And Reproducibility:**

The writing is clear, technical quality is good, the proposed method is novel and the code is provided as well.

**Strength And Weaknesses:**

+ The observation that correlated samples are labeled oppositely largely exists in contrastive learning. This is a fundamental issue and less touched in the literature. Compared to the prior SwAV and NNCLR, the proposed SNCLR adaptively mines correlated samples in the candidate neighbor set. The mining process is based on the cross-attention module which effectively measures the sample similarity during the loss computation phase. Both the samples and their positiveness are delved and formulated into the contrastive learning process.

+ Extensive experiments on standard benchmarks, with linear probe, semi-supervised training, transfer to detection and segmentations, with CNN and ViT backbones. Sota comparisons are performed as well. The overall writing is clear and the visualizations / figures are intuitive to represent the algorithm motivations and effectiveness.

-Compared to clustering and nearest neighbor selection, which are conducted in SwAV and NNCLR, the proposed SNCLR achieves better performance in the experiments. While claiming the combination of both advantages is an indirect way to illustrate the SNCLR effectiveness, more analysis shall be taken on elucidating why SNCLR performs better, especially using cross-attentions.

-Introducing more terms in Eq. 2 are shown to improve the performance. Any other alternatives to substitute current configurations (e.g., cross-attentions, weighted combination of terms)?


**Summary Of The Paper:**

A contrastive learning based self-supervised visual representation learning method is proposed in this paper. Upon observing the positive and negative assignments of training samples, this paper proposes to construct a candidate neighbor set where top candidate neighbors are selected as positive samples. When contribution of these positive samples are measured via cross attentions and defined as positiveness. These samples are denoted as soft neighbors to constitute the contrastive loss computation process. Experiments on several benchmarks and evaluation protocols indicate the feature representations of the proposed SNCLR perform well.

**Summary Of The Review:**

Overall, this is a good exploration on mining positive samples to support the contrastive learning computation. Sufficient experiments demonstrate the SNCLR is effective on both CNN and ViT encoders via various training settings and datasets.

---

### Official Review · Reviewer_v9QH · 2022-10-24

**Confidence:** 4
**Correctness:** 3
**Technical Novelty And Significance:** 2
**Empirical Novelty And Significance:** Not applicable
**Recommendation:** 6

**Clarity, Quality, Novelty And Reproducibility:**

Clarity. As mentioned above, the paper is well-written and organized. Some of the minor issues can be quickly fixed

Quality. The major idea makes sense. The whole paper is in a nice shape.

Novelty. The proposed method is extended from the existing works of NNCLR.

Reproducibility. The paper includes implementation details in the git repo.

**Strength And Weaknesses:**

Strength

The paper is clearly written and well organized.
Although the major idea is extended from the existing works of NNCLR, it still need to be merited.
The experiments are well conducted and the details are clear.

Weakness
1. The difference between equation 2 and the common contrastive loss is the NN extension in both Positive and Negative terms. In the ablation study, it is better to show the individual benefit of the NN. For example, in three experiments: a NN extension is only applied on the Positive. b NN extension is only applied on the Negative c NN extension is applied on both.
2. Since the performance gain in SSL is tend to be saturated, it's better to provide a standard deviation of the accuracy number for some gap which is less than 1% gain.
3. In section 3.2, "We have empirically tried using different structures of this attention module, including using parametric linear project.." comparing to complex computation, how about did some positive mining, a common method used in Deep Metric Learning [i]. which may be more efficient?

4. Some minor issues in the paper
Section 3.1
"For other N instance.." should be N-1 instance.
"We can also use only positive samples and substitute y1 with z1.." please define z1 in advance.


[i]Xuan, H., Stylianou, A., & Pless, R. (2020). Improved embeddings with easy positive triplet mining. In Proceedings of the IEEE/CVF Winter Conference on Applications of Computer Vision (pp. 2474-2482).

**Summary Of The Paper:**

This paper proposed a soft neighbor contrastive learning method (SNCLR) in the self-supervised learning(SSL) area. By pointing out the harshness of the binary positive and negative discrimination of the contrastive loss, the author introduces the soft positive by sampling the soft neighbor concept with the positiveness value. As the result, extensive experiments support the new method to achieve state-of-the-art results in the public benchmarks.


**Summary Of The Review:**

Overall, the paper is in good shape. Please take a look at the weakness.

---

### Official Review · Reviewer_ZPKB · 2022-10-25

**Confidence:** 3
**Correctness:** 4
**Technical Novelty And Significance:** 4
**Empirical Novelty And Significance:** 4
**Recommendation:** 8

**Clarity, Quality, Novelty And Reproducibility:**

Clarity: the paper is written well and easy to understand.

Quality: the paper is in a good quality with sufficient evaluations following the standard benchmarks.

Originality: using soft neighbors is new in contrastive learning.

**Strength And Weaknesses:**

Strength

+Using soft neighors is new in contrastive learning. The idea is well motivated and makes a lot sense.

+The experimental evaluations are sufficient and shows better performance than SOTA

Weakness

-There is no analysis on the memory complexity of the algorithm because a large pool of neighborhood set should be maintained

-How does the size of the neighborhood set affect the performance?

-The NNCLR result reported in Table 1 (a) is 74.5, while the original NNCLR paper reports 75.6. What is the reason behind that?

**Summary Of The Paper:**

This paper proposes to improve self-supervised contrastive learning by leveraging soft neighbours. The main idea is to maintain a candidate neighborhood set and retrieve K nearest neighbors for each sample as its positive examples. A positiveness score is calculated using an attention module and used as a soft weight of the neighbor. Experimentally, they show the proposed method improves the feature representations for many downstream tasks.

**Summary Of The Review:**

This paper proposes to leverage soft neighbors to improve contrastive learning, which is new and important.  The experiment also shows promising results. Therefore, I would recommend an accept.

---

### Decision · Program_Chairs · 2023-01-20

**Decision:**

Accept: poster

**Justification For Why Not Higher Score:**

While novel enough, the idea is not entirely new, which is reflected in the reviewers' novelty/significance ratings.

**Justification For Why Not Lower Score:**

No reviewers recommend rejection.

**Metareview: Summary, Strengths And Weaknesses:**

The paper proposes to go beyond the binary distinction of samples (pos/neg) and leverage correlated non-positive instances (soft neighbors) for representation learning. The reviewers appreciate the idea, writing and results. They request more analyses, metrics, baselines and ablations, but the findings of this paper merit publication even in the current form.

**Note From Pc:**

if the above contains the word "oral" or "spotlight" please see: "oral" presentation means -> notable-top-5% and "spotlight" means -> notable-top-25%. As stated in our emails, we are disassociating presentation type from AC recommendations